# The Response of Naturally Based Coatings and Citrus Fungicides to the Development of Four Postharvest Fungi

**DOI:** 10.3390/jof10050309

**Published:** 2024-04-24

**Authors:** Lizette Serrano-Molina, Mónica Hernández-López, Dolores Azucena Salazar-Piña, Silvia Bautista-Baños, Margarita de Lorena Ramos-García

**Affiliations:** 1Facultad de Nutrición, Universidad Autónoma del Estado de Morelos, Calle Iztaccihuatl S/N, Col. Los Volcanes, Cuernavaca 62350, Morelos, Mexico; lizette.serrano@uaem.edu.mx (L.S.-M.); azucena.salazar@uaem.mx (D.A.S.-P.); 2Centro de Desarrollo de Productos Bióticos, Instituto Politécnico Nacional, Carretera Yautepec-Jojutla, km. 6.8, Calle CEPROBI 8, Col. San Isidro, Yautepec 62731, Morelos, Mexico; mohernandezl@ipn.mx

**Keywords:** *Citrus* essential oils, commercial products, *Licopersicon esculentum* mill., mycotoxins, storage behavior, tomato

## Abstract

The tomato (*Licopersicon esculentum* Mill.) is considered to be one of the products with the highest demand due to its nutritional value; however, it is susceptible to infection by fungi during its pre- and postharvest stages. In this research, three commercial products (1% Citrocover, 1% Citro 80, and 0.002% Microdyn) and two coatings based on 1.0% chitosan/0.1% lime or 0.1% orange essential oils were evaluated in vitro and on Saladette tomatoes that were previously inoculated with four postharvest fungi. The application of the commercial citrus-based product Citrocover was highly effective in reducing the in vitro development of *Aspergillus flavus*, *Fusarium oxysporum*, and *Colletotrichum gloeosporioides*, but not *Rhizopus stolonifer*. The sanitizer Microdyn promoted infections with most fungi. Citrus-based products were effective in reducing infections with *A. flavus* in the tomatoes during storage. Overall, mycotoxin production was very low for all treatments. The use of commercial citrus-based products and coatings did not alter the weight loss, firmness, or total soluble solid contents of the treated tomatoes. The changes observed were, rather, associated with the normal ripening process of Saladette tomatoes. The commercial citrus-based products satisfactorily controlled the in vitro growth of the fungi *Aspergillus flavus*, *Fusarium oxysporum*, and *Colletotrichum gloeosporioides*.

## 1. Introduction

The tomato is a perishable and delicate vegetable; therefore, it is susceptible to bruises and perforations in its skin caused by improper handling at the time of harvest, which makes it susceptible to infection by microorganisms in the various stages of the postharvest chain. In 2016, in Mexico, the Food and Agricultural Organization [1] reported estimated rates of food loss and waste for tomatoes of all cultivated varieties of approximately 10.3% during harvest and 7.6% at different postharvest stages prior to the point of sale. During the postharvest stages, among the main microorganisms that cause damage affecting the quality of tomatoes are the fungi *Aspergillus flavus*, *Fusarium oxysporum*, *Colletotrichum gloeosporioides*, and *Rhizopus stolonifer*. The first two produce secondary metabolites called mycotoxins, which can be very toxic to consumers [2,3].

Controlling fungi in tomatoes has the purpose of extending their shelf life; generally, such strategies are applied to reduce and inhibit the production of spores and their germination, as well as the development of their mycelia and, consequently, the synthesis of mycotoxins [4]. Synthetic fungicides are the most commonly used in the control of postharvest fungi [5,6]. However, in spite of their favorable results in controlling various pathogens, their use is becoming increasingly questionable, which is mainly because their persistent and indiscriminate use has generated toxic waste and health effects; in addition, they lack effectiveness when fungal strains develop resistance, which makes it necessary to research new fungicides [7,8]. For this reason, it is necessary to evaluate products with fungicidal activity that do not cause negative effects on health.

The plant and essential oil extracts from the *Citrus* genus are generally rich in vitamin C, anthocyanins, and flavonoids, and they have high antimicrobial activity [9,10]. In addition, in combination with chitosan (a product resulting from the deacetylation of chitin obtained from the exoskeletons of insects and crustaceans) [11], they have shown great antifungal activity in phytopathogenic fungi in vitro and in situ. Some examples of this activity are provided in the following. According to Narvaez et al. [12], the best results for the inhibition of radial growth and spore production in *R. stolonifer* and *C. gloeosporioides* were obtained with the essential oils of sweet orange (*C. sinensis*) and tangerine (*C. nobilis*) obtained from leaves at 4%. In further experiments, among nine spices, including lime (*C. limon*), the production of aflatoxins was reduced to 60.6% after 1 h of treatment in naturally contaminated maize [13]. In addition, aqueous and ethanolic extracts of lime peels were tested at concentrations ranging from 62.5 to 500 mg/mL and were found to inhibit the radial growth and spore production of *F. oxysporum* [10].

Liu et al. [14] reported that aqueous and ethanolic extracts of mandarin orange showed the greatest inhibition of mycelial growth in *A. flavus* at concentrations from 300 to 400 mg mL^−1^. Similarly, commercial mandarin essential oil has shown reductions in *Penicillium digitatum* and *P. italicum* at concentrations of 40 and 50 ppm, reaching 100% inhibition in the case of *P. digitatum* [15].

On the other hand, coatings based on a combination of chitosan and citrus extracts reinforced their fungicidal activity when tested on tomatoes and mangoes. For example, a coating based on sweet orange essential oil at a concentration of 100 µL mL^−1^ with chitosan (2%) remarkably controlled the fungal decay caused by *A. niger* and *P. citrinum* in tomatoes during eight days of storage at 25 °C [16]. In another experiment carried out by Cheng et al. [17], pomelo extract (*C. grandis*) at concentrations of 10 and 20 mL^−1^ with the incorporation of chitosan suppressed anthracnose during the storage of mango cv ‘Keitt’, and the storage temperature was found to be another important factor for the reduction in anthracnose in the fruit.

Colloidal silver is characterized by its antibacterial action, especially in the inhibition of coliform bacteria in leaves and vegetables such as strawberries and cilantro [18,19]. For fungal pathogens, other commercial products such as Medicer colloidal silver and Craft colloidal silver have shown marked control of various fungi according to the concentrations applied in vitro. In this case, the most susceptible fungi to Craft colloidal silver was *Sclerotinia sclerotiorum* at 7.5 ppm, followed by *Alternaria brassica* and *Botrytris cinerea* at 15 ppm [20].

The objectives of this research were to compare the effectiveness of three commercial products and two coatings based on 1.0% chitosan/0.1% lime or 0.1% orange essential oils in relation to the in vitro development of four postharvest fungi, the incidence of disease, and the ripening process in Saladette tomatoes during daytime storage, as well as the production of mycotoxins (aflatoxins and fumosins). In addition, the ripening process of the treated tomatoes was evaluated.

## 2. Materials and Methods

### 2.1. Selection of Fungal Strains

The *A. flavus* fungus was obtained from the fungal collection of the Laboratory of Postharvest Technology of Agricultural Products at CEPROBI-IPN (San Isidro, Mexico). This strain was activated in tomato fruit and incubated in Czapeck-dox agar medium (BD Bioxon, Mexico City, Mexico) at 20 °C. The remaining three fungal strains were isolated from diseased tomatoes showing different symptomatologies. Molecular identification was carried out by the Laboratorio Nacional de Investigación y Servicios Agroalimentarios y Forestales at Chapingo Autonomous University in the state of Mexico, Mexico. The resulting fungi were *Colletotrichum gloeosporioides*, *Fusarium oxysporum*, and *Rhizopus stolonifer*. For these fungi, potato dextrose agar (BD Bioxon, Mexico City, Mexico) was used as a nutrient medium, and they were incubated at 20 °C.

### 2.2. Commercial Products

Three commercial products were evaluated: 1% Citrocover, 1% Citro-80 (both from MS Agros S.A. de C.V., Yautepec, Mexico), and 0.0021% Microdyn (Mercancías Salubres S.A. de C.V., Mexico City, Mexico). For 1% Citrocover and 1% Citro-80, 5 mL of the product was added to 500 mL of running water and mixed. For Microdyn, 2.5 mL of water was added and mixed.

### 2.3. Chitosan Essential Oil Coatings

#### 2.3.1. Chitosan Obtention

First, 1% low-molecular-weight chitosan (Q) (Sigma Aldrich CAS 9012-76-4, Mexico City, Mexico) (degree of deacetylation of 75–85%) was prepared by adding an equal amount (*w*/*v* 1:1) of acetic acid (Fermont Chemicals Inc., Monterrey, Mexico) to chitosan. This was stirred for 24 h. The solution was adjusted to pH 5.5 with a 1 N NaOH solution. The chitosan–acetic acid mixture was added to 500 mL of distilled water [21,22].

#### 2.3.2. Essential Oil Obtention

Press extraction was used to isolate the lime essential oil (LEO) and orange essential oil (OEO). The citrus peels were punctured to break the rind. Then, they were mechanically pressed to squeeze out the oils and juices. Once the oil was released, it was collected in a container, and the juice and oils were centrifuged to separate the liquid from the solids. For both essential oils, the final concentration used was 0.1% [23].

#### 2.3.3. Coatings

The above-mentioned 1% chitosan solution was stirred for 24 h. Then, 0.3% glycerol (J.T. Baker^®^, Mexico City, Mexico) and 0.1% essential oils (orange essential oil/lime essential oil) (*v*/*v*) were added. A homogenizer (Virtis, New York, NY, USA) was used at 13,500 rpm for 1 min.

### 2.4. Treatments

The treatments used in this research were the following: (a) 1% Citrocover, (b) 1% Citro 80, (c) chitosan (Q) + 0.1% orange essential oil (OEO) + 3% glycerol, and (d) 1.0% chitosan (Q) + 0.1% lime essential oil (LEO) + 3% glycerol and control (distilled water).

### 2.5. Tomato Fruit

Saladette tomatoes were obtained from orchards located in the municipality of Taltizapan (18°24′36″ N 99°04′13″ O) in the state of Morelos, Mexico. The fruit maturity index was full red (80 to 90% of the surface of the fruit had a red color). They were free of bruises, rots, and pests.

### 2.6. Antifungal Analysis

For these experiments, the variables considered were mycelial growth, spore germination, and disease development. For the in vitro variables, the methodology proposed was that of Cortes-Higareda et al. [21]. For this, 25 mL of each treatment was uniformly added to ten Petri plates (90 × 15 mm in diameter) containing Czapek-Dox or PDA medium. Petri dishes without treatments were used as a control. After drying, 10 μL of a spore suspension with a concentration of 10^5^ CFU/mL from each fungus was placed in the center of the Petri plate, allowed to dry, and incubated at 25 ± 2 °C.

The radial mycelial growth of the fungus for each treatment was measured every day with a Truper Vernier caliper for 7 days. The area was calculated using the formula a = *r*^2^ and expressed as mycelial growth (cm^2^).

For the conidial germination variable, 10 mL of sterile water was added to five Petri dishes that belonged to each treatment. After, conidia were scraped off the agar of each treatment with the aid of a glass rod. The number of spores/mL of the filtrate was adjusted to 10^5^ UFC using a Neubauer hemocytometer. From this spore suspension, aliquots of 30 μL were placed onto six PDA disks with a diameter of 5 mm. Germination was terminated by adding lactophenol–safranin. One hundred observations were conducted per treatment using a Nikon ALPHAPHOT-2YS2-H optical microscope with a 40× objective. For this variable, observations were carried out in 0, 2, 4, 6, 8, and 10 h incubation periods. The results are expressed as the percentage of spore germination.

For disease incidence, the tomatoes were prepared as follows: They were immersed for 15 s in a 2% hypochlorite solution, rinsed in running water, and dried at ambient temperature. Then, they were punctured 3 mm deep with a sterile needle and inoculated with 20 μL of a spore suspension at a concentration of 10^5^ UFC/mL. The same fungi and treatments that were evaluated in vitro were evaluated on the tomatoes. Each fungus was tested in separate experiments. The treatments were applied 24 h after inoculation by immersing the tomatoes in each formulation and they were stored at an ambient temperature of 25 °C. Each treatment was applied to 10 fruits with three repetitions.

The disease incidence variable was evaluated daily by measuring the mycelial growth of the fungi on the tomato with a digital Vernier both longitudinally and transversally. Measurements were taken at 4 days for *R. stolonifer*, 6 days for *F. oxysporum* and *C. gloeosporioides*, and 8 days for *A. flavus*. Data are expressed as cm^2^.

### 2.7. Mycotoxin Production In Vitro and on Inoculated Tomatoes

The aflatoxin and fumonisin production of *A. flavus* and *F. oxysporum*, respectively, was also measured in the treatments that were evaluated in vitro and on the inoculated tomatoes. In the latter case, tomatoes that were previously inoculated with *A. flavus* or *F. oxysporum* were dehydrated with a hot air flow with a food dehydrator (Hamilton Beach Commercial^®^, Richmond, VA, USA). Then, they were ground in a food processor (Oster^®^, Saltillo Coah, Mexico) and stored in a dry place for subsequent evaluations.

For the aflatoxin production, the methodology followed was that reported by Neogen Corporation. For this, 10 g of *A. flavus* incubated on Czapek-Dox or dehydrated samples was mixed with 50 mL of 70% methanol. The extract was filtered (Whatman paper No. 1) and kept in borosilicate tubes. Then, 100 µL of the conjugate (aflatoxin + monoclonal antibody supplied by the kit) was pipetted into a multiwell and mixed. Next, 100 µL of the calibration standard solution and the extract were transferred into antibody-coated wells and incubated for 10 min. They were rinsed five times with deionized water. Then, 100 µL of the substrate solution K-Blue^®^ was added, incubated for 10 min, and agitated; 100 µL of the stop solution Red Stop^®^ was added. Lastly, the mixed solution was placed into the Stat-Fax 4700 reader (Neogen Corporation, Lansing, MI, USA) at 650 nm. The results were expressed in ppb.

For fumonisins, the quantitative lateral-flow immunoassay test was considered. For this, 10 g of the sample was mixed with 50 mL of 65% ethanol and centrifuged at 1000 RPM (Gusto^®^, Dawson, IL, USA) for 1 min. The extract was reserved in Eppendorf tubes. Then, 200 µL of extract and 400 µL of diluent were added to the mixing tubes and homogenized. The cartridges with the reagent strips were placed in the Raptor^®^ Integrated Analysis Platform (Neogen Corporation, Lansing, MI, USA), and 400 µL of the mixture was added. The reading was performed automatically and is expressed in ppm.

### 2.8. Ripening of Treated Tomatoes

The weight loss, firmness, and total soluble solids (TSSs) were assessed according to the methodology of Aparicio-García et al. [24]. For weight loss, 5 tomatoes per treatment with three replicates were weighed every day for 7 days with a scale (OHAUS, Tokyo, Japan). Data are expressed as the percentage weight loss. For firmness, 3 fruits per treatment with two replicates were considered. The value was determined using an analogous penetrometer (KANDPI, Tokyo, Japan). Measurements were taken on both sides of the fruit. This variable was assessed at the end of the experiment. The values are reported in Newtons (N). To determine the TSSs, a drop of juice was extracted from 3 tomatoes and placed in a refractometer (Atago, Tokyo, Japan), and the value was taken. The results are expressed in °Brix.

### 2.9. Statistical Analysis

The experiments were arranged in a completely randomized design. An analysis of variance (ANOVA) and a Tukey means test (*p* ≤ 0.05) were then performed by using the statistical package in InfoStat student version 2017. Standard deviations were also calculated.

## 3. Results

### 3.1. In Vitro Growth and Disease Development on Inoculated Tomatoes

For *A. flavus*, the treatment with 1% Citrocover was the one that most inhibited its growth until the end of its 8-day storage, with a final growth of approximately 11.2 cm^2^, followed by the treatments with 1% Citro-80 with 28 cm^2^. With the remaining treatments, *A. flavus* had an average growth of approximately 45.0 cm^2^ (Figure 1a). When tomatoes were dipped in these same treatments, the infection was noticeably lower in the tomatoes treated with 1% Q/0.1% OEO (Figure 1b). In this case, the final value of the fungal development was approximately 3.5 cm^2^. In the case of the application of 0.0021% Microdyn, the growth of A. flavus was promoted in comparison with the remaining treatments and the control (8.7–10 cm^2^), since, at the end of storage, the approximate value was 14.6 cm^2^.

In relation to *C. gloeosporioides*, 1% Citrocover significantly (*p* < 0.05) inhibited the mycelial growth during the whole period, unlike the remaining treatments (Figure 2a). At the end, for the corresponding value, the growth was 1.2 cm^2^, while in the other treatments, the final average growth was from 43 to 50 cm^2^. On the other side, except for 0.0021% Microdyne, the growth of *C. gloeosporioides* on the inoculated tomatoes was minor in all treatments, including the control, during the storage period (Figure 2b). In general, the growth range was from approximately 1.7 to 3.8 cm^2^.

For *F. oxysporum*, 1% Citrocover also showed the strongest effect, followed by the 1% Q/0.1% LEO coating, since, at the end of the incubation period of 8 days, the growth achieved was only 3.78 and 15 cm^2^, respectively, which was statistically different (*p* < 0.05) among the treatments (Figure 3a). In the treated tomatoes, except for those treated with 0.0021% Microdyn, it was observed that, at the end of the 6-day storage period, the infection was significantly similar (*p* < 0.05) among the treatments, including the control (Figure 3b). In the case of 0.0021% Microdyne, the infection was significantly greater (16.2 cm^2^) and was statistically different from the remaining treatments.

Regarding the in vitro growth of *R. stolonifer*, the treatment in which the fungus showed the least development (20 cm^2^) at the end of the four-day incubation period was that with 1% Q/0.1% OEO (Figure 4a). In the remaining treatments, including the control, the mycelial growth was 50 and 60 cm^2^ (Figure 4a and Figure 5). Concerning the development of *R. stolonifer* in the inoculated tomatoes, regardless of the treatment applied, it advanced rapidly as the days of storage increased, without a marked difference compared to the control, except for 0.0021% Microdine. In this case, the growth of *R. stolonifer* was the lowest, with a corresponding value of 26 cm^2^ (Figure 4b and Figure 6). Statistical differences (*p* < 0.05) were observed among the treatments.

### 3.2. Spore Germination

The germination of *A. flavus* spores in Czapek-Dox medium was statistically different (*p* < 0.05) among the treatments. For the 1% Citrocover treatment, germination was reduced by approximately 60% at the end of the 10 h of incubation (Figure 7a). In relation to the 1% Citro-80 and 1% Q/0.1% OEO treatments, the inhibition was approximately 40%. The spore inhibition for *A. flavus* under the 0.0021% Microdyne treatment and control was very similar; for the latter, it was promoted. The development of *F. oxysporum* spores was similar in most treatments, including the untreated samples, at the end of the 10 h of incubation.

For *C. gloeosporioides*, the spore germination was significantly (*p* < 0.05) controlled among the treatments. Except for the treatment with 1% Q/0.1% LEO, the germination was inhibited from 60% to 80% with 10 h of incubation (Figure 8a). For this fungus, the 1% Citrocover and 1% Citro-80 products exerted the highest levels of inhibition. The inhibition of the germination of *R. stolonifer* spores was minimal, since, at the end of the 10 h incubation period, the spores germinated at 100% (Figure 8b). Only the treatment with 1% Citrocover partially inhibited the germination by 30% until the fourth day of incubation.

### 3.3. Mycotoxin Production In Vitro and on Inoculated and Dehydrated Tomatoes

Regarding the production of aflatoxin and fumonisins by *A. flavus* and *F. oxysporum*, statistical differences were observed among the treatments (*p* < 0.05) in the in vitro and in situ experiments (Table 1).

In the in vitro experiments, the lowest aflatoxin production was obtained with the commercial 1% Citrocover and 1% Citro-80 products, with a corresponding value of 0.1 ppb. In the in situ experiments, the production of aflatoxins in treated and untreated tomatoes was between 1.3 and 1.7 ppb. The lowest aflatoxin production was in tomatoes treated with the 1% Citrcover treatment, and the highest value corresponded to those treated with 0.0021% Microdyn.

With respect to the fumonisin production, the lowest in vitro production corresponded to the 0.0021% Microdyn (0.10 ppb) treatment, while the highest production corresponded to the control (0.21 ppm). The production of fumonisins in tomatoes treated with 1% Citro-80 (0.19 ppm) was the lowest compared to the rest of the treatments. The tomatoes coated with 1% Q/0.1% LEO showed the highest production (0.72 ppm).

### 3.4. Storage Behavior of the Treated Tomatoes

In general, the weight loss of tomatoes increased during the 7 days of storage in all treatments (Figure 9a). At the end of this storage period, it was observed that there were no significant differences (*p* < 0.05) between 1% Citrocover and the control. The greatest weight loss was evidenced in the tomatoes treated with 0.0021% Microdyne (10%), followed by those treated with 1% Q/0.1% LEO and 1% Q/OEO (8.7% and 7.2%, respectively). The lowest weight loss was observed in the tomatoes treated with 1% Citro-80 (5.9%).

There were significant differences (*p* < 0.05) in the firmness among the treatments (Figure 9b). However, except for that in the 0.00215 Microdyne treatment, the firmness was very similar. In general, the firmness values during the 7 days of storage were in the range from 2.1 N to 3.3 N. Overall, the highest values of firmness in the tomatoes corresponded to the 0.0021% Microdyne treatment.

For the TSSs, the values obtained were statistically different throughout the tomato storage period and across the treatments (Figure 10). There was no pattern in the evolution of the TSSs among the treatments and storage days, but in the case of 0.0021% Microdyne, the TSS content of the treated tomatoes was lower from the first to the last day of storage when compared with the remaining treatments, including the control.

## 4. Discussion

In this research, differences in fungal growth among treatments were found when measured on incubated Petri plates or in tomatoes. Likewise, the response to the treatments was different according to the genus of the fungus evaluated.

Except for the fungus *R. stolonifer*, the action of the citrus extracts—especially the commercial product 1% Citrocover—had a notable inhibitory effect in both stages of development in vitro, i.e., the mycelial development and spore viability of the remaining phytopathogenic fungi.

These results agree with those of previous in vitro studies. For example, Rodríguez and Montilla [25] mentioned that the commercial product Citrex, based on seed and pulp extracts of *C. paradisi*, had a significant effect on the mycelial development of *F. oxysporum f.* sp. *lycopersici*. Similarly, Segura-Palacios et al. [26] also evaluated 1% Citrocover in *A. flavus* isolated from tomatoes, obtaining a notably greater inhibitory effect than that of the control on mycelial growth and sporulation, with corresponding values of 3.0 cm^2^ and 0.2 × 10^5^ spores mL^−1^, respectively.

To date, various mechanisms of action against microorganisms (mainly fungi and bacteria) of citrus extracts have been described [27,28]. It has been mentioned that the union of the polyphenols of extracts with ergosterol, glycolipids of the bilayer, and proteins of the membrane of the pathogenic microorganism through covalent or hydrogen bonds resulted in their destabilization and thinning, thus generating a change in the balance of H+ and K+ and causing membrane rupture and a loss of cellular content [29].

In this research, we saw that the production of aflatoxins and fumonisins by *A. flavus* and *F. oxysporum*, respectively, in vitro and on dry tomatoes was very low, even in the control. In the case of aflatoxins, the synthesis at the end of storage was in the ranges from 0.1 to 0.5 ppb and from 1.3 to 1.7 ppb (in vitro and in situ, respectively), while the production of fumonisins was from 0.10 to 0.21 ppm and from 0.19 to 0.72 ppm (in vitro and in situ, respectively). Similar results were reported by Villegas-Rascón et al. [30] with the use of chitosan nanoparticles and encapsulated essential oils of cinnamon, thyme, and eucalyptus, which had no effects on the in vitro production of aflatoxins by *A. parasiticus*. On the other hand, in this research, it was observed that the concentration of mycotoxins in dehydrated tomatoes was slightly higher than that in in vitro studies. This could have been due to the composition of the matrix evaluated, since, in the in vitro studies, the growth of the fungi was based on a culture medium containing approximately 95% water, while in the case of the dehydrated tomatoes, there was a water content of approximately 5%. During the extractions for the immunoassays, a powder with a higher solid content than that of the nutrient medium could have resulted in a higher concentration of mycotoxins.

Nevertheless, the in situ results were not consistent with the in vitro ones or with previous findings that highlighted the effectiveness of citrus-extract-based fungicides in various agricultural products (watermelon, mango, and tomato) [10,16,17,31]. In the present research, the effect of the citrus-based fungicides was not significant in the tomatoes during storage; compared with the control (tomatoes sprayed only with water), only the application of 1% Q/0.1% OEO slightly reduced the development of *A. flavus*. As for *F. oxysporum* and *C. gloeosporioides*, the infection was very similar to that in the control for the citrus-based treatments, but it was stimulated with the application of 0.0021% Mycrodin. This is a commercial disinfectant made from colloidal silver and is indicated to prevent the development of bacteria and fungi. It has been described that the mechanism of action of colloidal silver is related to a cytotoxic function in the cell membrane of microorganisms, and it damages genetic material because free silver ions have an affinity for lipids and proteins. Therefore, they adhere to organic matter very easily [32]. However, scientific studies have reported the low effectiveness of colloidal silver in controlling microorganisms. Rangel-Vargas et al. [18] evaluated the effects of colloidal silver on 13 types of bacteria in pieces of jalapeño pepper; however, colloidal silver did not show significant differences compared with the control (saline solution). Similarly, Venat et al. [20] evaluated the effect of Medicer colloidal silver on pathogenic fungi in plants. The authors reported that colloidal silver resulted in 0% inhibition of mycelial growth in *A. flavus*, *A. niger*, and *Penicillium digitatum* at concentrations of up to 0.015 mL^−1^. Plata [33] described that microorganisms have a resistance to silver that can be caused by the release of energy-dependent ions from the cell by membrane proteins that function as ATPases.

Overall, the greatest infection at the end of tomato storage was caused by *R. stolonifer*, since it reached almost 75% of their surface area. The reason for this could be associated with the harvest maturity (full red) of the evaluated tomatoes. In this case, their ripening state could have caused the treated tomatoes to lose their ability to activate their resistance process and, hence, be able to prevent fungal infections by the inoculated fungi [34].

Concerning the three physicochemical variables measured on the tomatoes, except for the treatment with 0.0021% Microdyn, overall, the changes recorded could be associated with normal aspects of the final ripening process. In this regard, López-Vidal et al. [35] mentioned changes during the ripening of Saladette tomatoes through a loss of firmness and an increase in TSSs. In addition, in previous trials to evaluate the shelf life of tomatoes coated with 1% chitosan/0.1% lime essential oil, it was determined that the magnitude of these postharvest changes depends mainly on factors such as the type of coating, the ripening state, and the storage temperature [36].

## 5. Conclusions

The application of the commercial citrus-based product 1% Citrocover was highly effective in reducing the in vitro development of *A. flavus*, *F. oxysporum*, and *C. gloeosporioides*; this was not the case for *R. stolonifer*. However, the efficacy of 1% Citrocover and the remaining citrus-based fungicides was lower in all of the tested fungi, since they were evaluated in completely infected fruits.

The 1% Q/0.1% OEO coating was effective in reducing the in vitro development of *R. stolonifer*.

Except for the 0.0021% Microdyn sanitizer, the use of the commercial citrus-based products did not alter the ripening process of the treated tomatoes. The changes in the losses of weight, firmness, and the TSSs were associated with the normal ripening process of the tomatoes rather than the treatments applied.

## Figures and Tables

**Figure 1 jof-10-00309-f001:**
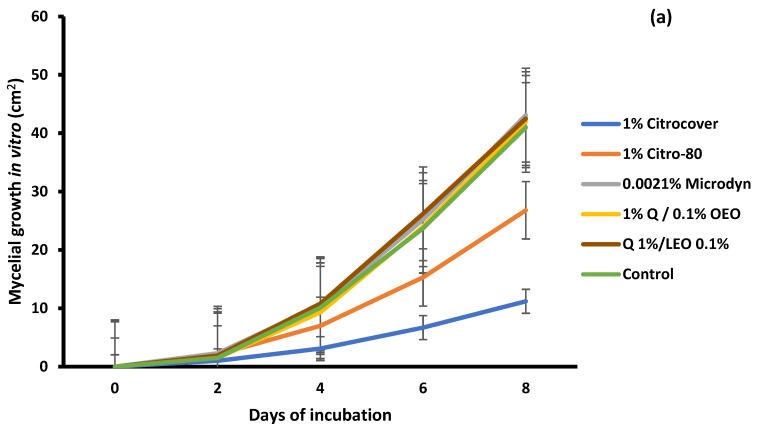
Evaluation of the mycelial growth of *Aspergillus flavus* treated with three commercial fungicides and two naturally based coatings under (**a**) in vitro conditions and on (**b**) Saladette tomatoes with an 8 h incubation time and 7 days of storage, respectively, at 25 ± 2 °C. Bars indicate the standard deviation of the mean. Q = chitosan, OEO = orange essential oil, and LEO = lime essential oil.

**Figure 2 jof-10-00309-f002:**
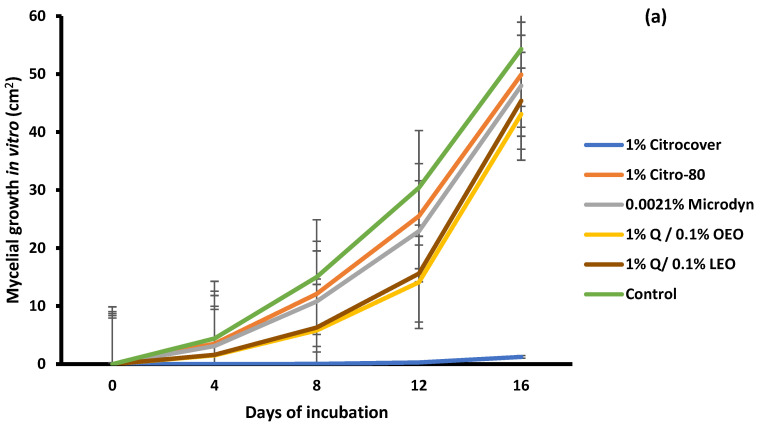
Evaluation of the mycelial growth of *Colletotrichum gloeosporioides* treated with three commercial fungicides and two naturally based coatings under (**a**) in vitro conditions and on (**b**) Saladette tomatoes with a 16 h incubation time and 6 days of storage, respectively, at 25 ± 2 °C. Bars indicate the standard deviation of the mean. Q = chitosan, OEO = orange essential oil, and LEO = lime essential oil.

**Figure 3 jof-10-00309-f003:**
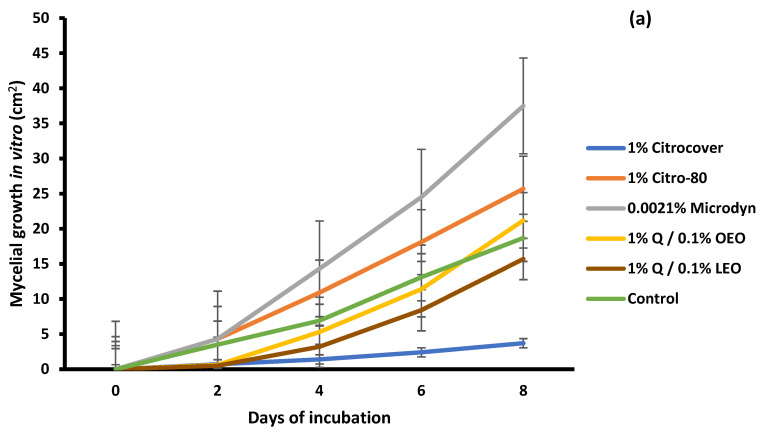
Evaluation of the mycelial growth of *Fusarium oxysporum* treated with three commercial fungicides and two naturally based coatings under (**a**) in vitro conditions and on (**b**) Saladette tomatoes with an 8 h incubation time and 6 days of storage, respectively, at 25 ± 2 °C. Bars indicate the standard deviation of the mean. Q = chitosan, OEO = orange essential oil, and LEO = lime essential oil.

**Figure 4 jof-10-00309-f004:**
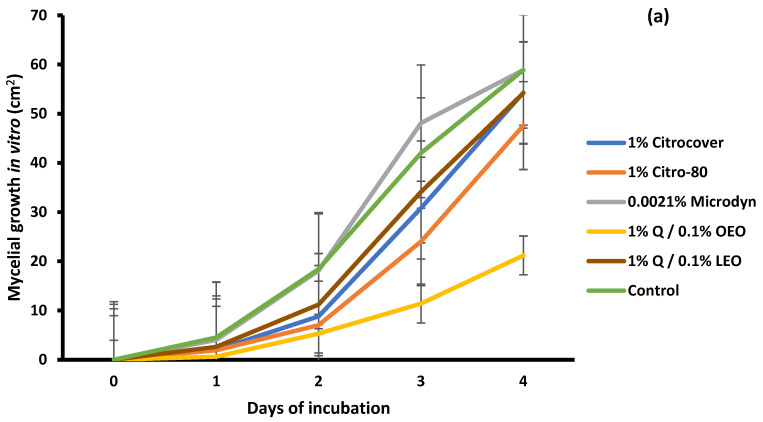
Evaluation of the mycelial growth of *Rhizopus stolonifer* treated with three commercial fungicides and two naturally based coatings under (**a**) in vitro conditions and on (**b**) Saladette tomatoes with a 4 h incubation time and 3 days of storage, respectively, at 25 ± 2 °C. Bars indicate the standard deviation of the mean. Q = chitosan, OEO = orange essential oil, and LEO = lime essential oil.

**Figure 5 jof-10-00309-f005:**
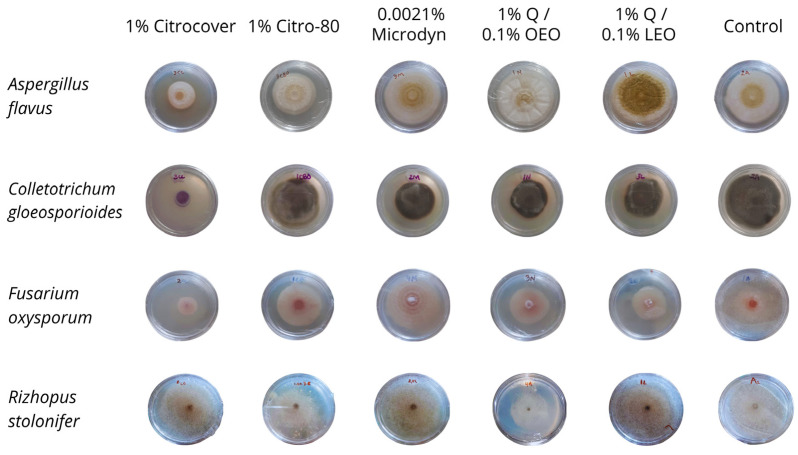
Mycelial growth of four postharvest fungi treated with three commercial fungicides and two naturally based coatings under in vitro conditions. Q = chitosan, OEO = orange essential oil, and LEO = lime essential oil.

**Figure 6 jof-10-00309-f006:**
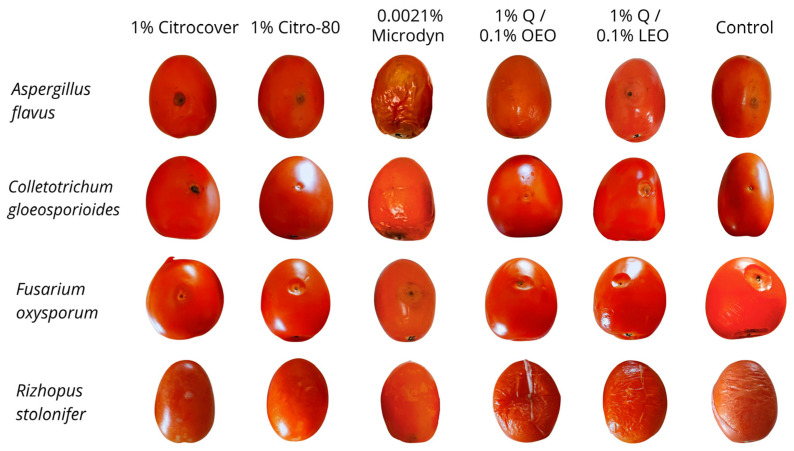
Mycelial growth of four postharvest fungi treated with three commercial fungicides and two naturally based coatings on Saladette tomato. Q = chitosan, OEO = orange essential oil, and LEO = lime essential oil.

**Figure 7 jof-10-00309-f007:**
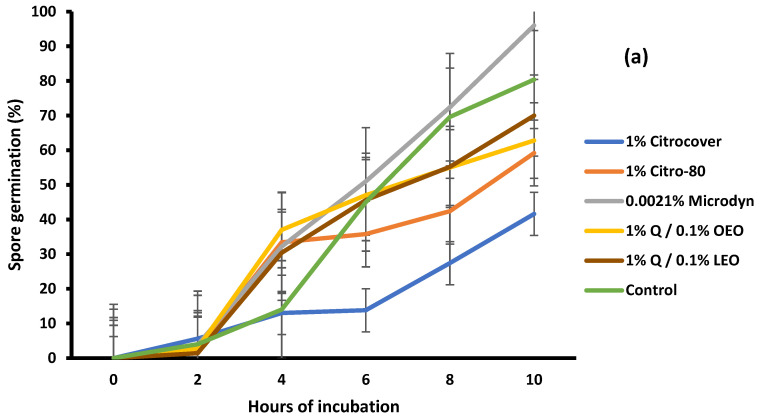
Spore germination of (**a**) *Aspergillus flavus* and (**b**) *Fusarium oxysporum* when incubated on PDA amended with three commercial fungicides and two naturally based coatings for 10 h at 25 ± 2 °C. Bars indicate the standard deviation of the mean. Q = chitosan, OEO = orange essential oil, and LEO = lime essential oil.

**Figure 8 jof-10-00309-f008:**
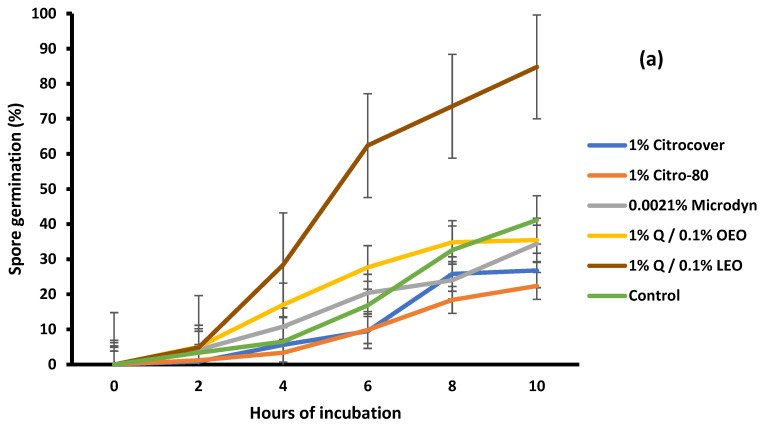
Spore germination of (**a**) *Colletotrichum gloeosporioides* and (**b**) *Rhizopus stolonifer* when incubated on PDA amended with three commercial fungicides and two naturally based coatings for 10 h at 25 ± 2 °C. Bars indicate the standard deviation of the mean. Q = chitosan, OEO = orange essential oil, and LEO = lime essential oil.

**Figure 9 jof-10-00309-f009:**
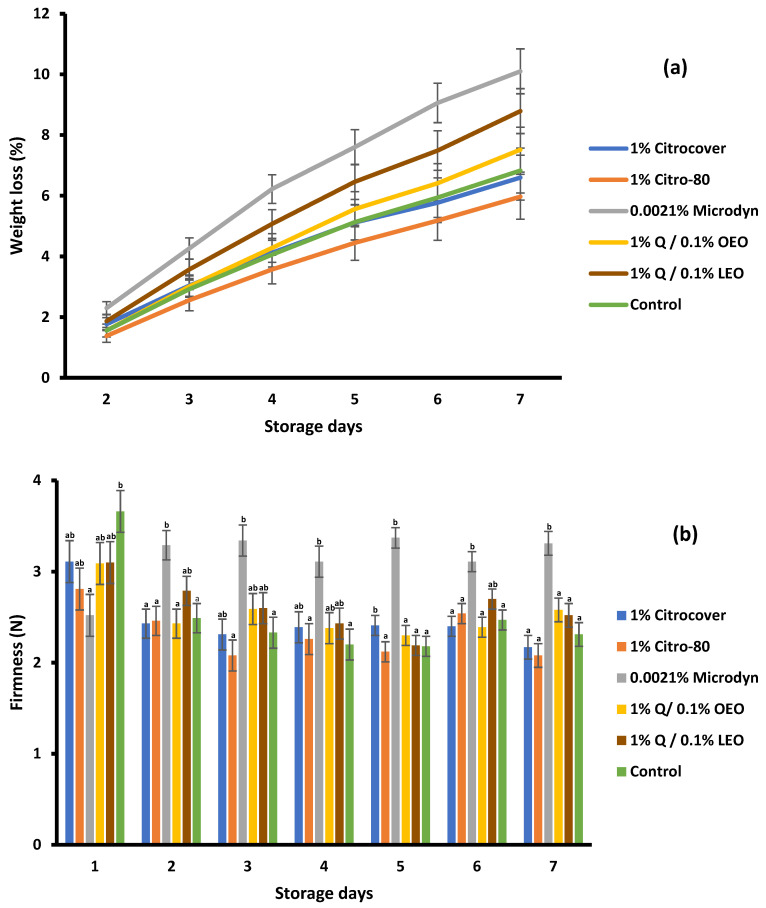
(**a**) Weight loss and (**b**) firmness of Saladette tomatoes treated with three commercial fungicides and two naturally based coatings over 7 days of storage at 25 ± 2 °C. Bars indicate the standard deviation of the mean. Q = chitosan, OEO = orange essential oil, and LEO = lime essential oil.

**Figure 10 jof-10-00309-f010:**
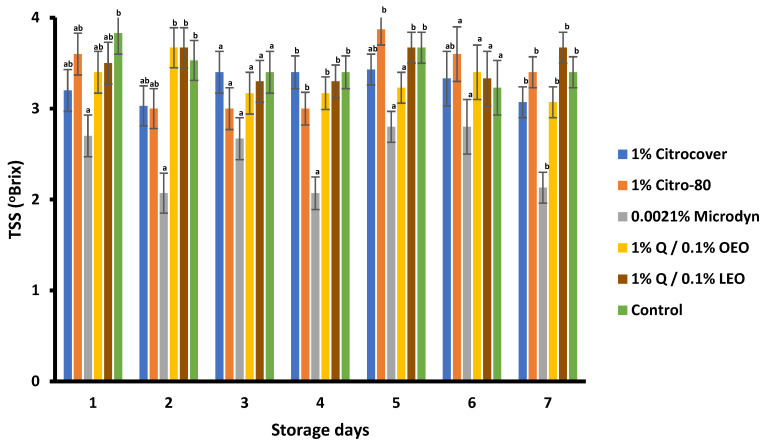
TSS content of Saladette tomatoes treated with three commercial fungicides and two naturally based coatings over 7 days storage at 25 ± 2 °C. Bars indicate the standard deviation of the mean. Q = chitosan, OEO = orange essential oil, and LEO = lime essential oil.

**Table 1 jof-10-00309-t001:** Evaluation of the production of aflatoxin and fumonisins by *Aspergillus flavus* and *Fusarium oxysporum*, respectively, in vitro and on tomatoes.

Treatments	Fungi			
	*Aspergillus flavus*		*Fusarium oxysporum*	
	In Vitro (ppb)	Dry Tomatoes (ppb)	In Vitro (ppm)	Dry Tomatoes (ppm)
1% Citrocover	0.1 ^a,^*	1.3 ^a^	0.16 ^b^	0.19 ^a^
1% Citro-80	0.1 ^a^	1.4 ^a^	0.15 ^b^	0.66 ^c^
0.002% Microdyn	0.3 ^b^	1.7 ^b^	0.10 ^a^	0.56 ^b^
1% Q/0.1% OEO	0.4 ^c^	1.4 ^a^	0.18 ^b^	0.43 ^c^
1% Q/0.1% LEO	0.2 ^a^	1.6 ^b^	0.18 ^b^	0.72 ^d^
Control	0.5 ^d^	1.6 ^b^	0.21 ^b^	0.67 ^c^

* Means followed by the same letter are not significantly different (*p* < 0.05) as determined with Tukey’s multiple-range test.

## Data Availability

Data are contained within the article.

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
