# Peer review of "The Response of Naturally Based Coatings and Citrus Fungicides to the Development of Four Postharvest Fungi"

_jof, 2024, doi:10.3390/jof10050309_

Round 1
Reviewer 1 Report
In the manuscript, Serrano-Molina et al evaluated the efficacy of various commercial citrus-based products and chitosan-based natural citrus extracts in inhibiting postharvest fungi on tomatoes. Citrocover 1% demonstrated high effectiveness against several fungi, particularly Aspergillus flavus, while citrus+chitosan-based coatings did not show significant impact on fungal growth, with minimal alterations observed in tomato quality parameters. Of particular interest are the findings regarding the effectiveness of the commercial vegetable wash Microdyn 0.0021%. The authors noted that Microdyn appeared to promote infection compared to other commercial products, which is interesting.
The discussion needs to be elaborated, and the rationale of the findings should be explained to strengthen the paper. For example, the authors found Microdyn at 0.0021% promoted the infection, compared to other commercial products. It would be good to compare and discuss the active ingredients or the concentration of the active ingredients of the three commercial products used in the study.
1. The authors should include the treatment used for controls in the materials and methods section. For example, in line 160 authors state that “For this, to ten Petri plates (90 x 15 mm of diameter) containing Czapek-Dox or PDA medium, 25 ml of each treatment were uniformly added”. How did they treat the control in this case?
2. Line 162 – “After drying, 10 ml of 10 μl of a spore suspension of 105/ml concentration of each fungus were placed in the center of the Petri plate, let to dry, and incubated at 25 ± 2 ºC.” I assume that the authors inoculated 10 μl in the Petri dish. I think 10 ml mentioned is a typing error.
3. Line 176 – “they were rapidly immersed in a 2% hypochlorite solution”. It would be good to give the time such as 15 secs or 30 secs, instead of using the term “rapidly immersed”.
4. Fig 7 – indicate Fig a and b in the legend. The fig 7b is confusing. The Y-axis title is"'firmness", and the graph looks like the firmness increases for Microdyn 0.0021% from day 2 -7. But it is actually “firmness loss”. To clarify the interpretation of Figure 7b, it would be beneficial to adjust the legend to explicitly state that higher values on the Y-axis represent increased firmness loss. This adjustment will ensure readers understand that the graph reflects a decrease in firmness over time for Microdyn 0.0021%.
5. This manuscript requires language editing before accepting for publication. Although some paragraphs are fairly well written, I recommend revising the manuscript with the assistance of a native English speaker.
6. Authors, please ensure to provide the full forms of any abbreviations used.
a. Line 43 - FAO
b. Line 214 – TSS
7. Keep consistency in the units' format (Ex: ml or mL).
Author Response
Manuscript ID: jof-2940815
Type of manuscript: Article
Response to Reviewer 1 Comment
Thank you very much for taking the time to review this manuscript. Please find the responses below and the corresponding corrections highlighted in the re-submitted file.
Comments 1: The discussion needs to be elaborated, and the rationale of the findings should be explained to strengthen the paper. For example, the authors found Microdyn at 0.0021% promoted the infection, compared to other commercial products. It would be good to compare and discuss the active ingredients or the concentration of the active ingredients of the three commercial products used in the study.
Response 1: The mechanism of action is complemented in the discussion part.
Comments 2: The authors should include the treatment used for controls in the materials and methods section. For example, in line 160 authors state that “For this, to ten Petri plates (90 x 15 mm of diameter) containing Czapek-Dox or PDA medium, 25 ml of each treatment were uniformly added”. How did they treat the control in this case?
Response 2: The information was added to the document.
Comments 3: Line 162 – “After drying, 10 ml of 10 μl of a spore suspension of 105/ml concentration of each fungus were placed in the center of the Petri plate, let to dry, and incubated at 25 ± 2 ºC.” I assume that the authors inoculated 10 μl in the Petri dish. I think 10 ml mentioned is a typing error.
Response 3: Correction was made to the text. However, in the English edition the texts were modified
Comments 4: Line 176 – “they were rapidly immersed in a 2% hypochlorite solution”. It would be good to give the time such as 15 secs or 30 secs, instead of using the term “rapidly immersed”.
Response 4: Correction was made to the text
Comments 5: Fig 7 – indicate Fig a and b in the legend. The fig 7b is confusing. The Y-axis title is"'firmness", and the graph looks like the firmness increases for Microdyn 0.0021% from day 2 -7. But it is actually “firmness loss”. To clarify the interpretation of Figure 7b, it would be beneficial to adjust the legend to explicitly state that higher values on the Y-axis represent increased firmness loss. This adjustment will ensure readers understand that the graph reflects a decrease in firmness over time for Microdyn 0.0021%.
Response 5: The table title was corrected
Comments 6: This manuscript requires language editing before accepting for publication. Although some paragraphs are fairly well written, I recommend revising the manuscript with the assistance of a native English speaker.
Response 6: The document was sent to English Editing services at MDPI
Comments 7: Authors, please ensure to provide the full forms of any abbreviations used.
- Line 43 - FAO
- Line 214 – TSS
Response 7: Correction was made to the text
Comments 8: Keep consistency in the units' format (Ex: ml or mL).
Response 8: Correction was made to the text

Reviewer 2 Report
The manuscript investigated the effect of the three commercial products Citrocover 1%, Citro 80 1%, and Microdyn 0.002%, and two coatings based on chitosan 1.0% / lime 0.1% or orange 0.1% essential oils on the inhibitory the development of the pathogens (four postharvest fungi), mycotoxins accumulation in vitro and in vivo, finally, compared the changes of weight loss, firmness and TSS after treatment. However, the whole level does not meet the level of the Journal.
General comments
1. The Introduction is too much, the author should rewrite it.
2. The design of the study is not quite reasonable, why the author selected the four postharvest fungi, please explain.
3. The experimental method is not clearly described, eg. how to obtain the coating?
4. The discussion part is need to rewrite, and added the inhibitory mechanism.
5. The author should supply some pictures on inhibitory on PDA or tomato fruit.
1. For Figure 1b, Figure 2b, Figure 3b, the scale of the y-axis should change, they are suitable proportion.
2. p<0.05 should be italic.
3. All the figures have no scale mark.
Author Response
Manuscript ID: jof-2940815
Type of manuscript: Article
Response to Reviewer 2 Comment
Thank you very much for taking the time to review this manuscript. Please find the responses below and the corresponding corrections highlighted in the re-submitted file.
Comments 1: The Introduction is too much, the author should rewrite it.
Response 1: The introduction was shortened.
Comments 2: The design of the study is not quite reasonable, why the author selected the four postharvest fungi, please explain.
Response 2: The postharvest fungi were selected because they are the fungi present in postharvest in tomato. This information is described in the introduction.
Comments 3: The experimental method is not clearly described, eg. how to obtain the coating?
Response 3: Correction was made to the text.
Comments 4: The discussion part is need to rewrite, and added the inhibitory mechanism.
Response 4: Mechanisms of action added to the discussion.
Comments 5: The author should supply some pictures on inhibitory on PDA or tomato fruit.
Response 5: Two figures from the in vitro and tomato experiments were added.
Comments 6: For Figure 1b, Figure 2b, Figure 3b, the scale of the y-axis should change, they are suitable proportion.
Response 6: It was modified and the same scale was used for similar variables.
Comments 7: p<0.05 should be italic.
Response 7: Correction was made to the text.
Comments 8: All the figures have no scale mark.
Response 8: Correction was made

Reviewer 3 Report
This manuscript by Margarita de L. Ramos-García et al. In vitro and In situ Response of Based-Natural Coatings and Biofungicides to the Development of Four Postharvest Fungal. To the end, this manuscript showed that the mycotoxin production was very low for all treatment. The use of the citrus-based products and coatings did not alter the weight loss, firmness, and total soluble solids content of the treated tomatoes. The changes observed were rather associated with the normal ripening process of Saladette tomatoes.
Nevertheless, in order to be accepted for publication, this manuscript needs major revision.
一、 Title
(1) There is a grammar error. “Based-Natural” should be modified to “Natural - Based”
(2) The description is too broad and does not specify which coating and biocide were used.
(3) Fungal" is an adjective that should be modified to its plural noun form: "Fungi."
(4) The entire text revolves around the study of tomatoes, yet the title does not mention tomatoes. “Four Postharvest Fungal” should be modified to”Four Tomato Postharvest Fungi”
二、Abstract
(1) Concentration symbols are not placed before the material names, for example, chitosan 1.0% and Citrocover 1% should be changed to 1.0% chitosan and 1% Citrocover.
(2) Overall, the mycotoxin production was very low for all treatment. In this sentence, the singular and plural usage is incorrect; "all treatment" should be changed to "all treatments."
(3) The use of the citrus-based products and coatings did not alter the weight loss, firmness, and total soluble solids content of the treated tomatoes. "The citrus-based products” should correspond to the preceding 'commercial products' in the text; suddenly changing the word may confuse the reader.
(4) keywords first letters are not capitalized.
(5) The last sentence should be a summary and reflection.
三、Introduction
(1)The first two, in addition produce secondary metabolites called mycotoxins, which can be very toxic to the consumer.——The sentence does not correctly use punctuation. A comma should be added after 'in addition'.
(2)"their persistent and indiscriminate use have generated toxic waste and affectations to health" should be changed to "their persistent and indiscriminate use has generated toxic waste and health effects"。
(3)Sentences:31-50——The first three paragraphs are intended to describe the current situation in a certain field. The length should not be too long; it should be concise and to the point. Additionally, the initial sentences do not use the keywords from the title, and irrelevant phrases are present. The title is focused on four kinds of post-harvest fungi, while the introductory part starts with a lengthy description of tomatoes. Introduction to four common fungi in tomatoes, as well as natural coatings and biofungicides, has not been mentioned.
(4) Sentences:51-62——The preceding sentence before "However" should be appropriately shortened or placed in the preceding paragraph.
(5)Sentences:63-66——This paragraph should be placed in the front section. After introducing what issues exist, this paragraph can be mentioned.
(6)Sentences:67-109——The following paragraphs discuss specific research findings from literature, which are too detailed. They should be summarized in a few sentences and integrated into the preceding paragraphs.
(7)Sentences: 110-115——“two coatings-based Citrus spp essential oils” should be changed to” two coatings based on 1.0% chitosan / 0.1%lime or 0.1%orange essential oils”
四、Materials and Methods
(1)Sentences:137-138——The phrase "1.0%" is already sufficient as a unit of concentration; the word "concentration" should not appear again.
(2)Sentences:141-142——In the sentence, '500 ml' is already sufficient as a unit of volume, so the word 'volume' should not appear again.
(3)Sentences:144-148——The sentence is chaotic and the meaning is unclear; it should be modified to: Press extraction was used for isolating the lime essential oil (LEO) and orange essential oil (OEO). The citrus peels were punctured to break the rind. After, they were mechanically pressed to squeezed out the oils and juices. Once the oil was released, it was collected in a container where the juice and oils were centrifuged to separate the liquid from the solids. For both essential oils, the final concentration used was 0.5%.
(4)Sentences:150-152——Concentration symbols are not placed before the material names.
(5)Sentences:155-156——This sentence has grammatical errors. “(80 to 155 90% of the surface fruit red color) ”should be modified to“(80 to 90% of the surface of the fruit is red in color)“
(6)Sentences:160-161——The sentence is not smooth. “to ten Petri plates (90 x 15 mm of diameter) containing Czapek-Dox or PDA medium, 25 ml of each treatment were uniformly added. “should be modified to” 25 ml of each treatment were uniformly added to ten Petri plates (90 x 15 mm of diameter) containing Czapek-Dox or PDA medium.
(7)Sentences:162——The unit is incomplete. “105 / ml” should be modified to “105CFU / ml”
(8)“In vitro fungal development and on the treated tomatoes“should be modified to” Antifungal analysis“,The title should be concise and to the point.
(9)The Materials and Methods section does not specifically specify several types of treatments and what is used as a control.
(10)There is no space before the first sentence of each paragraph.
The above only lists some of the inappropriate parts in the text. There are numerous problems with the content in the materials and methods section, especially with many sentences being incoherent and serious instances of word misplacement. Particularly, there is a tendency to elaborate unnecessarily on concepts that could be succinctly explained with a single unit symbol, resulting in a lack of conciseness. Some subheadings also contain grammar errors, making it difficult for readers to understand smoothly when translated.
五、Results
(1)There is no space before the first sentence of each paragraph.
(2)The names of the test fungi are not labeled in several of the pictures.
(3)The length of each subheading is too long, lacking conciseness, and there are errors in both syntax and grammar.
(4)Concentration symbols are not placed before the material names.
(5) The table title does not place the key word at the beginning. “Aflatoxin and fumonisins production by Aspergillus flavus and Fusarium oxysporum, respectively, in in vitro evaluations and on tomatoes.” should be modified to“evaluations of Aflatoxin and fumonisins production by Aspergillus flavus and Fusarium oxysporum, respectively, in in vitro and on tomatoes.”
(6) The explanatory text below Figure 7 and Figure 8 does not indicate which one is "a" and which one is "b".
The above only lists some of the inappropriate parts in the text. There are still many sentences that are not smooth in the content. In addition, each subheading is not concise and to the point, making it difficult for readers to understand. It would be helpful to set up a separate subheading for each measured indicator.
六、Discussion
Sentences:371——Try to use active voice as much as possible in the discussion section. “It was observed differences in the fungal growth when it was measured on incubated Petri plates or in the tomatoes among treatments.” should be modified to “We can see differences in the fungal growth when it was measured on incubated Petri plates or in the tomatoes among treatments.”
Sentences:381-384——The discussion section should not introduce new data; all data should come from the Results section.
The arrangement of the first four paragraphs and the final three paragraphs in the discussion section is well done, with the incorporation of others' findings to support one's own conclusions. However, the fifth and sixth paragraphs solely introduce others' results without presenting the research findings of this study, specifically regarding the production of fungal toxins in vitro and in vivo. Additionally, it would be advisable to include a brief paragraph at the end to elucidate some new questions and speculations based on your results.
七、Conclusions
(1)Concentration symbols are not placed before the material names.
(2) Only a few commercial products were summarized, and the effects of coatings were not summarized.
Author Response
Manuscript ID: jof-2940815
Type of manuscript: Article
Response to Reviewer 3 Comment
Thank you very much for taking the time to review this manuscript. Please find the responses below and the corresponding corrections highlighted in the re-submitted file.
Title
Comments 1: There is a grammar error. “Based-Natural” should be modified to “Natural - Based”
Response 1: Correction was made to the text. However, in the English edition the texts were modified
Comments 2: The description is too broad and does not specify which coating and biocide were used.
Response 2: Correction was made
Comments 3: Fungal" is an adjective that should be modified to its plural noun form: "Fungi."
Response 3: Correction was made
Comments 4: The entire text revolves around the study of tomatoes, yet the title does not mention tomatoes. “Four Postharvest Fungal” should be modified to”Four Tomato Postharvest Fungi”
Response 4: Correction was made
Abstract
Comments 5: Concentration symbols are not placed before the material names, for example, chitosan 1.0% and Citrocover 1% should be changed to 1.0% chitosan and 1% Citrocover.
Response 5: Correction was made
Comments 6: Overall, the mycotoxin production was very low for all treatment. In this sentence, the singular and plural usage is incorrect; "all treatment" should be changed to "all treatments."
Response 6: Correction was made
Comments 7: The use of the citrus-based products and coatings did not alter the weight loss, firmness, and total soluble solids content of the treated tomatoes. "The citrus-based products” should correspond to the preceding 'commercial products' in the text; suddenly changing the word may confuse the reader.
Response 7: Correction was made.
Comments 8: keywords first letters are not capitalized.
Response 8: Correction was made
Comments 9: The last sentence should be a summary and reflection.
Response 9: Correction was made.
Introduction
Comments 10: The first two, in addition produce secondary metabolites called mycotoxins, which can be very toxic to the consumer.——The sentence does not correctly use punctuation. A comma should be added after 'in addition'.
Response 10: Correction was made to the text. However, in the English edition the texts were modified.
Comments 11: "their persistent and indiscriminate use have generated toxic waste and affectations to health" should be changed to "their persistent and indiscriminate use has generated toxic waste and health effects"
Response 11: Correction was made.
Comments 12: Sentences:31-50——The first three paragraphs are intended to describe the current situation in a certain field. The length should not be too long; it should be concise and to the point. Additionally, the initial sentences do not use the keywords from the title, and irrelevant phrases are present. The title is focused on four kinds of post-harvest fungi, while the introductory part starts with a lengthy description of tomatoes. Introduction to four common fungi in tomatoes, as well as natural coatings and biofungicides, has not been mentioned.
Response 12: The introduction was modified.
Comments 13: Sentences:51-62——The preceding sentence before "However" should be appropriately shortened or placed in the preceding paragraph.
Response 13: Correction was made.
Comments 14: Sentences:63-66——This paragraph should be placed in the front section. After introducing what issues exist, this paragraph can be mentioned.
Response 14: This paragraph was removed in the change to the introduction.
Comments 15: Sentences:67-109——The following paragraphs discuss specific research findings from literature, which are too detailed. They should be summarized in a few sentences and integrated into the preceding paragraphs.
Response 15: The information was shortened.
Comments 16: Sentences: 110-115——“two coatings-based Citrus spp essential oils” should be changed to” two coatings based on 1.0% chitosan / 0.1%lime or 0.1%orange essential oils”
Response 16: Correction was made to the text. However, in the English edition the texts were modified.
Materials and Methods
Comments 17: Sentences:137-138——The phrase "1.0%" is already sufficient as a unit of concentration; the word "concentration" should not appear again.
Response 17: Correction was made.
Comments 18: Sentences:141-142——In the sentence, '500 ml' is already sufficient as a unit of volume, so the word 'volume' should not appear again.
Response 18: Correction was made.
Comments 19: Sentences:144-148——The sentence is chaotic and the meaning is unclear; it should be modified to: Press extraction was used for isolating the lime essential oil (LEO) and orange essential oil (OEO). The citrus peels were punctured to break the rind. After, they were mechanically pressed to squeezed out the oils and juices. Once the oil was released, it was collected in a container where the juice and oils were centrifuged to separate the liquid from the solids. For both essential oils, the final concentration used was 0.5%.
Response 19: Correction was made.
Comments 20: Sentences:150-152——Concentration symbols are not placed before the material names.
Response 20: Correction was made.
Comments 21 Sentences:155-156——This sentence has grammatical errors. “(80 to 155 90% of the surface fruit red color) ”should be modified to“(80 to 90% of the surface of the fruit is red in color)“
Response 21: Correction was made. However, in the English edition the texts were modified.
Comments 22: Sentences:160-161——The sentence is not smooth. “to ten Petri plates (90 x 15 mm of diameter) containing Czapek-Dox or PDA medium, 25 ml of each treatment were uniformly added. “should be modified to” 25 ml of each treatment were uniformly added to ten Petri plates (90 x 15 mm of diameter) containing Czapek-Dox or PDA medium.
Response 22: Correction was made.
Comments 23: Sentences:162——The unit is incomplete. “105 / ml” should be modified to “105CFU / ml”
Response 23: Correction was made.
Comments 24: “In vitro fungal development and on the treated tomatoes“should be modified to” Antifungal analysis“,The title should be concise and to the point.
Response 24: Correction was made.
Comments 25: The Materials and Methods section does not specifically specify several types of treatments and what is used as a control.
Response 24: Correction was made.
Comments 26: There is no space before the first sentence of each paragraph.
Response 26: Correction was made.
Comments 27: The above only lists some of the inappropriate parts in the text. There are numerous problems with the content in the materials and methods section, especially with many sentences being incoherent and serious instances of word misplacement. Particularly, there is a tendency to elaborate unnecessarily on concepts that could be succinctly explained with a single unit symbol, resulting in a lack of conciseness. Some subheadings also contain grammar errors, making it difficult for readers to understand smoothly when translated.
Response 27: The document was sent for English review at MPDI, to correct errors
Results
Comments 28: There is no space before the first sentence of each paragraph.
Response 28: Correction was made.
Comments 29: The names of the test fungi are not labeled in several of the pictures.
Response 29: All images indicate the fungus evaluated.
Comments 30: The length of each subheading is too long, lacking conciseness, and there are errors in both syntax and grammar.
Response 30: The titles of the graphics were modified.
Comments 31: Concentration symbols are not placed before the material names.
Response 31: Correction was made.
Comments 32: The table title does not place the key word at the beginning. “Aflatoxin and fumonisins production by Aspergillus flavus and Fusarium oxysporum, respectively, in in vitro evaluations and on tomatoes.” should be modified to“evaluations of Aflatoxin and fumonisins production by Aspergillus flavus and Fusarium oxysporum, respectively, in in vitro and on tomatoes.”
Response 32: Correction was made.
Comments 33: The explanatory text below Figure 7 and Figure 8 does not indicate which one is "a" and which one is "b".
Response 33: Correction was made.
Comments 34: The above only lists some of the inappropriate parts in the text. There are still many sentences that are not smooth in the content. In addition, each subheading is not concise and to the point, making it difficult for readers to understand. It would be helpful to set up a separate subheading for each measured indicator.
Response 34: The document was sent for English review at MPDI, to correct errors. The results are separated by a subheading for each indicator measured.
Discussion
Comments 35: Sentences:371——Try to use active voice as much as possible in the discussion section. “It was observed differences in the fungal growth when it was measured on incubated Petri plates or in the tomatoes among treatments.” should be modified to “We can see differences in the fungal growth when it was measured on incubated Petri plates or in the tomatoes among treatments.”
Response 35: Correction was made.
Comments 36: Sentences:381-384——The discussion section should not introduce new data; all data should come from the Results section.
Response 36: Correction was made.
Comments 37: The arrangement of the first four paragraphs and the final three paragraphs in the discussion section is well done, with the incorporation of others' findings to support one's own conclusions. However, the fifth and sixth paragraphs solely introduce others' results without presenting the research findings of this study, specifically regarding the production of fungal toxins in vitro and in vivo. Additionally, it would be advisable to include a brief paragraph at the end to elucidate some new questions and speculations based on your results.
Response 37: This section was completed.
Conclusions
Comments 38: Concentration symbols are not placed before the material names.
Response 38: Correction was made.
Comments 39: Only a few commercial products were summarized, and the effects of coatings were not summarized.
Response 39: Correction was made.

Round 2
Reviewer 2 Report
The author responsed all the comments and modfied them.
The author responsed all the comments and modfied them.
Reviewer 3 Report
Accept in present form
Accept in present form